# Use of Cardiac Contractility Modulation in an Older Patient with Non-Ischemic Dilated Cardiomyopathy: A Case Report

**Gianvito Manganelli [1], Antonio Fiorentino [2], Gianluca Ceravolo [2], Stefana Minichiello [1], Giuseppe Bianchino [1], Gennaro Bellizzi [1,†], Giuseppe Pacileo [3] and Daniele Masarone [3,*]**

[1]  Cardiology Department, ASL Avellino–Sant'Ottone Frangipane Hospital, 83031 Ariano Irpino, Italy; gianvito.m@teletu.it (G.M.); stefana@gmail.com (S.M.); bianchino@gmail.com (G.B.); gbellizzi58@gmail.com (G.B.)

[2]  Impulse Dynamics Germany, 60487 Frankfurt am Main, Germany; afiorentino@impulsedynamics.com (A.F.); ceravolo@gmail.com (G.C.)

[3]  Heart Failure Unit, Department of Cardiology, A. O. dei Colli, Monaldi Hospital, 80131 Naples, Italy; gpacileo58@gmail.com

*  Correspondence: danielemasarone@gmail.com; Tel.: +39-081-7065-163; Fax: +39-081-7062-674

†  This paper is dedicated to the memory of Gennaro Bellizzi, Chief of Cardiology Unit at Sant'Ottone Frangipane Hospital, who passed away while this paper was being peer-reviewed.

**Abstract:** Cardiac contractility modulation (CCM) is a novel device-based therapy used in patients with HFrEF. CCM therapy is associated with an improvement in exercise tolerance, increased quality of life, reduced HF hospitalizations, and reverse remodelling of the left ventricle in patients with HFrEF. In this case, we report the clinical benefit of CCM in an older patient with advanced HFrEF due to ischemic dilated cardiomyopathy with frequent heart failure-related hospitalizations and poor quality of life despite optimal medical therapy.

**Keywords:** cardiac contractility modulation; chronic heart failure; non-ischemic dilated cardiomyopathy; quality of life





## 1. Introduction

Despite significant advances in pharmacologic and nonpharmacologic therapy, many patients with heart failure and reduced ejection fraction (HFrEF) still have a clinical history characterized by frequent disease readmission, reduced quality of life, and poor prognosis [1,2]. In patients with HFrEF, current guidelines suggest optimized medical therapy (OMT) with disease-modifying drugs, and device therapy according to the patient's phenotype [3]. In patients with ischemic aetiology and persistence of a left ventricular ejection fraction (LVEF) <35% despite OMT, an implantable cardioverter–defibrillator (ICD) must be implanted to prevent arrhythmic sudden cardiac death (class I recommendation, A level evidence). However, more freedom is left to clinical judgment in patients with a non-ischemic aetiology (class IIa recommendation, A level evidence). In addition, a cardiac resynchronization therapy (CRT) device is used for patients with HFrEF and QRS $\geq$ 150 ms with a left bundle-branch block-type morphology (class I recommendation, A level evidence). However, only one-third of patients with HFrEF meet these criteria and 30% are non-responders after CRT implantation [4]. In patients with symptomatic HF despite OMT, LVEFs between 25% and 45%, and who are not eligible for CRT, randomized clinical trials and clinical registry have shown that cardiac contractility modulation (CCM) therapy represents a valid therapeutic option in improving exercise capacity and quality of life, in alleviating HF-related symptoms, and in reducing HF-related hospitalizations [5,6]. In this case, we show the use of CCM therapy in an older patient with dilated cardiomyopathy due to non-ischemic aetiology without indication to CRT, demonstrating the clinical benefit of this novel HF device-based therapy.

## 2. Case Report

A 78-year-old man was admitted to the cardiology department of Sant'Ottone Frangipani Hospital in October 2019 for acute HF. His past medical history includes resistant hypertension (since 2000), chronic kidney disease stage III (since 2013), and hypertensive cardiomyopathy in the dilated phase with a severe reduction in systolic function (LVEF 40%) since 2012. In addition, in 2006, the patient underwent an implant with a dual-chamber pacemaker (PM) because of brady–tachy syndrome. The PM programming was set in VVI mode at a 50 b/m base rate since 2012 due to permanent atrial fibrillation (AF), for which an ablation procedure was also proposed but refused by the patient. Since 2017, the patient had an LVEF of 30% and a history of frequent hospitalizations despite OMT (>2/years) with bisoprolol 7.5 mg/die, sacubitril/valsartan 49/51 mg bis in die (the patient developed hyperkalaemia with a $[K^+]$ of 5.8 mEq/L at a dosage of 97/103 mg bis in the die), and eplerenone 25 mg/die. During January 2019, a coronary angiography was performed with the absence of significant coronary artery disease. At clinical admission, the patient was in class NYHA III, with signs of peripheral congestion (leg oedema, elevated jugular venous pressure, and acute kidney injury) but without signs of hypoperfusion. The ECG showed atrial fibrillation with a mean ventricular response of 70 b/m, an end-diastolic volume of 152 mL, an end-systolic volume of 108 mL, and an LVEF of 30% (Figure 1). The echocardiography also showed grade III diastolic dysfunction at a medium degree of mitral and tricuspid regurgitation.

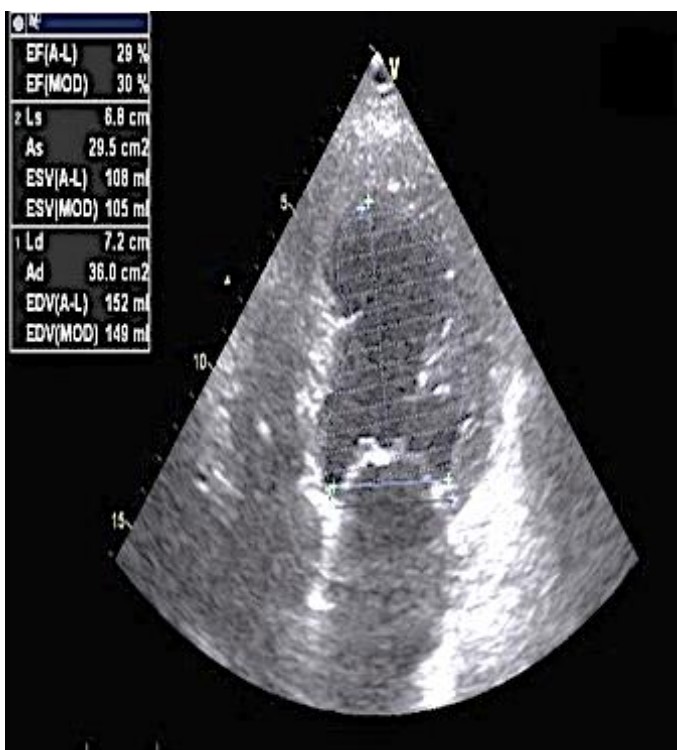

**Figure 1.** The four chambers viewed at admission. Note the severe reduction in systolic function (LVEF 30% using the Simpson biplane method).

The NT-proBNP value was 1305.4 pg/mL, and the Minnesota Living with Heart Failure Questionnaire (MLHFQ) score was 82 points. The patient was treated with intravenous diuretics and levosimendan and achieved a euvolemia state after three days of admission with a reduction in NT-proBNP to 976.8 pg/mL. According to ESC guidelines on the management of chronic and acute HF and the results of DANISH trials [5], given the low risk of arrhythmic sudden cardiac death of the patient, we did not opt for ICD implantation; instead, to reduce HF-related hospitalizations and to improve quality of life, we decided

to implant a CCM therapy device. After signing informed consent, the Optimizer Smart® (Impulse Dynamics, Marlton, NJ, USA) for the delivery of CCM therapy was implanted (Figure 2). During the implantation procedure, a total dose of 8 mL of iopromide was used, and due to the minimal amount used, no contrast-induced nephropathy occurred during follow-up.

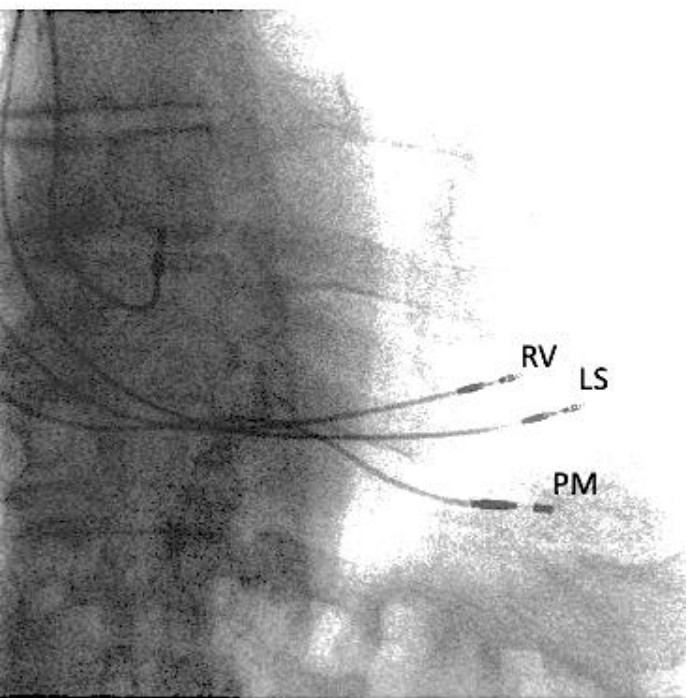

**Figure 2.** Fluoroscopy with two visible ventricular septal leads for CCM delivery (RV and LS) and a lead for ventricular pacing of the pacemaker (PM).

Therapy was initially scheduled for 8 h per day, with the impulse of the CCM delivered (Figure 3) only from the first lead (reference ventricular, RV) due to the presence of discomfort caused by the high-energy signal delivered from the second lead (local sense, LS), but no interference between PM and Optimizer Smart® occurred. Additionally, no increase in tricuspid regurgitation occurred after implantation of the two leads in the right ventricle for the delivery of CCM therapy.

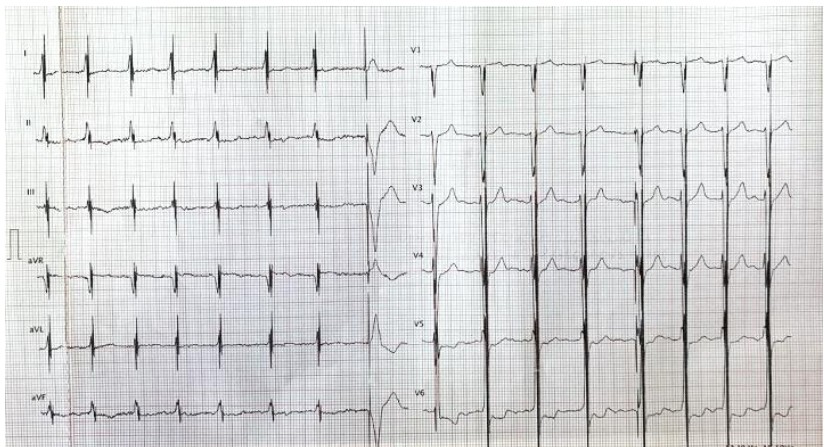

**Figure 3.** Electrocardiogram with visible spikes of high-energy impulses from CCM therapy.

At 3-month follow-up, the patient already experienced significant improvements in terms of QOL, with an MLWHFQ score of 11 and no HF-related hospitalizations. In addition, a decrease in BNP to 600 pg/mL was observed. However, despite clinical improvement, considering the low delivery rate of therapy (58.6%) due to a ventricular rate above 110 bpm (at rates above this, CCM therapy is inhibited), and despite optimal rate control therapy, the CCM therapy delivery time was increased to 12 h per day. At 18-month follow-up, no HF-related hospitalizations occurred and the patient had stable NYHA class II medical therapy, an LVEF of 41% (Figure 4), an MLWHFQ score of 6, an NT-proBNP value of 273.3 pg/mL, stable renal function, and a CCM therapy delivery rate of 84.3% (due to a persistent ventricular rate response of AF < 110 b/m).

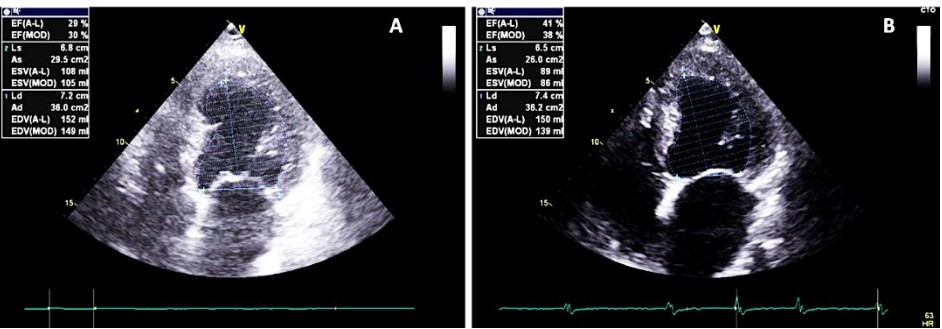

**Figure 4.** The four chambers viewed at baseline (**A**) and at 18-month follow-up (**B**). Note the reverse remodelling of the left ventricle with an increase in systolic function of the left ventricle (LVEF 41% using the Simpson biplane method).

## 3. Discussion

CCM is an innovative HF device therapy that acts through a high-energy, nonexcitatory biphasic signal in the absolute ventricular refractory period of the heart cycle delivered by two active fixation leads placed on the right interventricular septum [7]. The mechanism of action of CCM in HF patients is mainly based on the improvement in calcium handling and normalization of pathologic HF gene expression (expression of a foetal gene). Improvement in calcium handling occurred through the phosphorylation of phospholamban, which increased the calcium reuptake into the sarcoplasmic reticulum [8]. Additionally, the HF gene expression profile was found to be reversed (toward normal) in patients who received CCM [9]. These two mechanisms justify the improvement in cardiac contractility (in the absence of increased myocardial oxygen consumption) that CCM therapy should be used for patients with HFrEF [10]. CCM therapy was associated with improvements in exercise tolerance (both the distance covered in the 6-min walking distance test and the $VO_2$ peak in the cardiopulmonary exercise test), an increase in quality of life, a reduction in HF hospitalizations, and reverse remodelling of the left ventricle (reduction in left ventricular volumes and increase in LVEF) [5,11,12]. The current guidelines on the diagnosis and management of acute and chronic heart failure by the European Society of Cardiology [3] state that CCM should be considered in patients with HF, NYHA class III–IV HFs, an LVEF ≥25% to ≤45%, and a QRS duration <130 ms.

Considering our patient features (age >70 years, non-ischemic dilated cardiomyopathy, narrow QRS, NYHA class IIIb, and multiple HF-related hospitalizations), the risk of sudden arrhythmic death was much lower compared with the risk of death from other causes (mainly HF) [13,14]. Therefore, CCM therapy that ameliorates HF symptoms and reduces HF-related hospitalizations was preferred over an ICD that only prevents arrhythmic sudden cardiac death but has no effects on quality of life and HF-related hospitalizations. Additionally, at 18-year follow-up, patients have an LVEF of 41% and, therefore, an indication to ICD is not present anymore. In addition, in our patient, heart transplant was excluded due to the older age of the patient (the maximum age set at our unit is 70 years) and the implantation of a mechanical left ventricular assistance device

has a maximum age of 75 years in our unit. Therefore, in older patients with HFrEF who have NYHA class III-IV, frequent HF-related hospitalizations, and clinical features that suggest a low risk of arrhythmic sudden cardiac death, we believe that CCM therapy can be considered to improve functional capacity, to reduce hospitalizations, and to reverse remodelling of the left ventricle so that implantation of an ICD is no longer necessary.

## 4. Conclusions

CCM represents an emerging effective and safe HF device therapy for the treatment of HFrEF not eligible for CRT implantation. In the present report, the use of CCM therapy as a standalone device led to improved quality of life and a reduction in HF-related hospitalizations in an older patient with HFrEF due to non-ischemic dilated cardiomyopathy with symptoms refractory to OMT medical therapy and frequent HF-related hospitalizations. Therefore, as a medical treatment, device therapy (which in the next few years will consist of an increasing number of devices) should be tailored for each patient with HFrEF.

**Author Contributions:** Conceptualization: G.M., A.F., G.C. and G.P.; writing—original draft preparation: D.M. and S.M.; writing—review and editing: G.B. (Giuseppe Bianchino) and G.B. (Gennaro Bellizzi). All authors have read and agreed to the published version of the manuscript.

**Funding:** The APC was funded by Impulse Dynamics Germany GmbH.

**Institutional Review Board Statement:** Not applicable.

**Informed Consent Statement:** Informed consent was obtained from all subjects involved in the study.

**Data Availability Statement:** Not applicable.

**Acknowledgments:** We are grateful to Impulse Dynamics for their unconditional support in this project.

**Conflicts of Interest:** A.F. and G.C. are a full-time employee of Impulse Dynamics where Optimizer Smart® was discovered and developed.

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
