# Peer review of "Use of Cardiac Contractility Modulation in an Older Patient with Non-Ischemic Dilated Cardiomyopathy: A Case Report"

_clinpract, doi:10.3390/clinpract11040098_

Round 1

Reviewer 1 Report

In this interesting case report by Gianvito et al., the authors report the case of a patient with non-ischemic dilated CMP who was successfully treated with CCM implantation that reduced HF symptoms and hospitalization despite a background of optimal medical therapy.

The report is interesting, but the following issues should be addressed:

  1. Did this patient with multiple cardiovascular risk factors have an angiogram excluding significant CAD in light of his severe systolic disfunction? This point is further relevant as the Authors state that the patients was not offered an ICD in light of his non-ischemic cardiomyopathy. Please report on this point;
  2. As far as this reviewer can see there is no description of patient’s medication. One could wonder how much dose titration of HF drugs was done considering the stage 2 CKD. Was this patient given sacubitril/valsartan? Please address these points. There were any bystander comorbidities limiting the introduction of evidence based HF therapy such as hyperkalemia restricting ACE/ARB/MRAs use?
  3. What was the LVEF in the previous 2 years before CCM implantation?
  4. What was the pre-discharge NT-proBNP value after proper in-hospital diuretic and levosimendan cycle? This is relevant to define the true NT-proBNP delta;
  5. Did the patient experienced any discomfort or complication during the CCM implantation? How much contract media was used to complete the procedure? Did he have a transient acute kidney injury? Was the tricuspid valve injured in some ways with residual tricuspid insufficiency? Did you have any interference between CCM system and PM?
  6. Did the amelioration in LVEF seen after 18 months from initiation of CCM therapy translate in an improvement of the renal function allowing to introdutce/uptitrate HF medications? 
  7. Please consider to include a very recent review on CCM therapy which summarizes the main evidence in HF by leader opinions in this field (doi: 10.1714/3660.36450).

Author Response

In this interesting case report by Gianvito et al., the authors report the case of a patient with non-ischemic dilated CMP who was successfully treated with CCM implantation that reduced HF symptoms and hospitalization despite a background of optimal medical therapy

The report is interesting, but the following issues should be addressed:

Did this patient with multiple cardiovascular risk factors have an angiogram excluding significant CAD in light of his severe systolic dysfunction? This point is further relevant as the Authors state that the patients was not offered an ICD in light of his non-ischemic cardiomyopathy. Please report on this point;

Response: We thank the reviewer for this comment. We have specified that patient has performed a coronary angiography ten months (January 2019) before presentation at our Unit (October 2019)

As far as this reviewer can see there is no description of patient’s medication. One could wonder how much dose titration of HF drugs was done considering the stage 2 CKD. Was this patient given sacubitril/valsartan? Please address these points. There were any bystander comorbidities limiting the introduction of evidence-based HF therapy such as hyperkalemia restricting ACE/ARB/MRAs use?

We thank the reviewer for this comment. We have specified the therapy of the patient with disease-modifying drugs and that hyperkalemia limit further increase of sacubitril/valsartan

 What was the LVEF in the previous 2 years before CCM implantation?

Response: We thank the reviewer for this comment. We have specified the LVEF of the patient in the two years before the CCM implant.

What was the pre-discharge NT-proBNP value after proper in-hospital diuretic and levosimendan cycle? This is relevant to define the true NT-proBNP delta;

Response: We thank the reviewer for this comment. We have added in the text the NT-proBNP value after achieved of euvolemia was 976.8 pg/ml.

Did the patient experience any discomfort or complication during the CCM implantation? How much contract media was used to complete the procedure? Did he have a transient acute kidney injury? Was the tricuspid valve injured in some ways with residual tricuspid insufficiency? Did you have any interference between the CCM system and PM?

Response: We thank the reviewer for this comment. We have added in the text the dose of contrast agent used during the Optimizer Smart implantation; The patients don’t have transient acute kidney injury, no tricuspid valve injury occurred during implant and no inference between the two devices occurred. 

Did the amelioration in LVEF seen after 18 months from initiation of CCM therapy translate in an improvement of the renal function allowing to introduce/uptitrate HF medications? 

Response: We thank the reviewer for this comment. We have added in the text that CCM not induce an improvement of estimated glomerular filtration rate and no further optimization of medical therapy was performed during follow-up    

Please consider including a very recent review on CCM therapy which summarizes the main evidence in HF by leader opinions in this field (doi: 10.1714/3660.36450).

Response: We thank the reviewer for this comment. We have added the reference according to the reviewer suggestion.  

Reviewer 2 Report

Dear Authors.

This is an interesting case study of a successful treatment of an older patient with nonischemic heart failure with a reduced ejection fraction and multiple rehospitalizations with the use of a cardiac contractility modulation device (CCM).  In my opinion this case has a true didactic potential; however, it needs a more detailed presentation.

My main critical remarks are;

1 No data on pharmacological treatment were provided before the decision on GCM implantation and during follow-up. It is important to realize that the improvement of patient heart status can be multifactorial (medical treatment and device effects). The statement in the article about OMT is not enough for me.

  1. Data on echocardiogrpahic examination are rather sparse and oriented to the ejection fraction. Please provide more details on echocardiographic examination including data on mitral and tricuspid regurgitation -if present?
  2. In the discussion section more options should be addressed Only ICD vs. CGM was discussed. For non-cardiologists a discussion why this patient is not a good candidate for heart transplantation or mechanical circulatory support (LVAD). If a mitral or tricuspid incompetence is present, a discussion of mitraclip use or surgical intervention should be addressed.
  3. Discuss the current status of CCM therapy according to recent ESC guidelines should be discussed.
  4. Last but not least, a thorough overview of English grammar and spelling is needed in order to achieve an acceptable readability of some parts of the text.

Author Response

Dear Authors.

This is an interesting case study of a successful treatment of an older patient with nonischemic heart failure with a reduced ejection fraction and multiple rehospitalizations with the use of a cardiac contractility modulation device (CCM).  In my opinion this case has a true didactic potential; however, it needs a more detailed presentation.

My main critical remarks are:

No data on pharmacological treatment were provided before the decision on GCM implantation and during follow-up. It is important to realize that the improvement of patient heart status can be multifactorial (medical treatment and device effects). The statement in the article about OMT is not enough for me.

We thank the reviewer for this comment. We have specified the therapy of the patient with disease modifying drugs and that hyperkalemia limit further increase of sacubitril/valsartan.

Data on echocardiographic examination are rather sparse and oriented to the ejection fraction. Please provide more details on echocardiographic examination including data on mitral and tricuspid regurgitation -if present?

We thank the reviewer for this comment. We have added some echocardiographic data and clarify that no severe valvular regurgitations were present.

In the discussion section more options should be addressed Only ICD vs. CGM was discussed. For non-cardiologists a discussion why, this patient is not a good candidate for heart transplantation or mechanical circulatory support (LVAD). If a mitral or tricuspid incompetence is present, a discussion of mitraclip use or surgical intervention should be addressed.

We thank the reviewer for this comment. We have added in the discussion section a section on Htx and LVAD.

Discuss the current status of CCM therapy according to recent ESC guidelines should be discussed.

We thank the reviewer for this comment.

Unfortunately, the new guidelines on treatment and management of HF limit  CCM to a single phrase however we have added that "indication" in the text.

Last but not least, a thorough overview of English grammar and spelling is needed in order to achieve an acceptable readability of some parts of the text.

We thank the reviewer for this comment. We have sent the manuscript to a professional English editing.

Round 2

Reviewer 1 Report

The authors have adequately addressed my previous comments.

Author Response

The authors thanks the reviewer for their help in improve the manuscript

Reviewer 2 Report

Dear Authors

Thank you for the revision of your article.

Two small remarks -for minor corrections:

  1. in Introduction line 40 -There is a misunderstanding . you can not say that this patient is  a non responder to CRT. If I understood he has no indication to upgrade his double chamber (right atrial and right ventricular lead) to a CRT device due to narrow QRS complex. It is quite  a different story- please correct.

2 The meaning of the statement is difficult to understand :

line 340

"Additionally,at 18-yearfollow-up, patients have an LVEF of 41%and therefore the response to CRT is no longer there."

Please speciy if it is a citation of finding of some study -if so insert a reference or rephrase or delete  it.

Author Response

We thank the reviewer for their comments that allow us to correct two mistakes.

We have changed the text as follow 

1) "In this case, we show the use of CCM therapy in an older patient with dilated cardiomyopathy due to non-ischemic aetiology without indication to CRT, demonstrating the clinical benefit of this novel HF device-based therapy."

2)Additionally, at 18-year follow-up, patients have an LVEF of 41% and therefore indication to ICD is not present anymore